# *p*-Hydroxybenzoic Acid β-d-Glucosyl Ester and Cimidahurinine with Antimelanogenesis and Antioxidant Effects from *Pyracantha angustifolia* via Bioactivity-Guided Fractionation

**DOI:** 10.3390/antiox9030258

**Published:** 2020-03-21

**Authors:** Sun-Yup Shim, Ye Eun Lee, Hwa Young Song, Mina Lee

**Affiliations:** 1Fish Health Center, Chonnam National University, 50 Daehak-Ro, Yeosu-si, Jeonnam 59626, Korea; shimsy@scnu.ac.kr; 2College of Pharmacy, Sunchon National University, 255 Jungangno, Suncheon-si, Jeonnam 57922, Korea; qjsro1124@naver.com (Y.E.L.); blueocean33@nate.com (H.Y.S.)

**Keywords:** *Pyracantha angustifolia*, p-Hydroxybenzoic acid β-d-glucosylester, Cimidahurinine, melanin, tyrosinase, antioxidant

## Abstract

This study evaluated bioactivity-guided fractionation as a means to identify therapeutic phytochemicals from *Pyracantha angustifolia* that can attenuate melanogenesis and oxidation. Seven compounds with inhibitory effects on melanin production and tyrosinase (TYR) activity, and ABTS and DPPH radical-scavenging activities, which have not been reported as whitening materials, were isolated from the *n*-butanol fraction from *P. angustifolia* leaves (PAL). Among the seven compounds, *p*-hydroxybenzoic acid β-d-glucosylester (HG), and cimidahurinine (CH) had strong inhibitory effects on melanin production and TYR activity, as well as ABTS and DPPH radical-scavenging activities. Western blot analysis showed that HG and CH suppressed tyrosinase-related protein (TYRP)-1 and TYRP-2 expression. Moreover, HG and CH inhibited reactive oxygen species (ROS) generation in tert-butyl hydroperoxide (*t*-BHP)-treated B16F10 cells. These results suggest that *P. angustifolia* containing active compounds, such as HG and CH, is a potent therapeutic candidate for the development of hypopigmenting agents.

## 1. Introduction

Melanogenesis involves the production of melanin pigments in melanosomes produced by melanocytes [1,2]. Melanoblasts, the precursor cells of melanocytes, migrate to various of the body and develop into melanocytes, which are predominantly found in the basal layer of skin epidermis and hair follicles [3,4]. Melanin is important for the prevention of UV-induced DNA damage by binding reactive oxygen species (ROS) that form from exposure to ultraviolet (UV) radiation [5,6,7]. Melanogenesis is enhanced by the activation of tyrosinase (TYR), the key enzyme of melanogenesis. TYR is an enzyme that determines the color of mammalian skin and hair in the melanogenesis of mammals and browning of fruits and fungi [8]. Tyrosine is hydrozylated to 3,4-dihydroxyphenylalanin (DOPA) in melanin synthesis with TYR as a catalyst, which is a copper-containing enzyme present in melanosomes, and oxidized to dopaquinone by TYR [9]. Mammalian melanogenic enzymes are similar metalloproteins, such as TYR, tyrosinase-related protein 1 (TYRP-1), and TYRP-2. TYRP-1 and TYRP-2 are present in melanosomes and catalyze eumelanin-producing reactions. TYRP-1 increases the eumelanin: pheomelanin ratio and protects against oxidative stress through its peroxidase activity [10]. In contrast, TYRP-2 is the enzyme dopachrome tautomerase (DCT), which facilitates the tautomerization of l-DOPAchrome (a red melanin precursor) to colorless dihydroxyindole-2-carboxylic acid (DHICA). In the absence of TYRP-2, l-DOPAchrome is converted to dihydroxyindole (DHI), a toxic melanin precursor, which has a significant effect on the properties of the melanins produced [9,10,11,12]. Therefore, melanocytes might regulate the local and global homeostasis of the melanogenic system by controlling TYR, TYRP-1, and TYRP-2. Oxidative stress, which is defined as the production of reactive oxygen species (ROS) and toxic free-radical species, as well as an imbalance between pro-oxidant and antioxidant homeostasis, is associated with a range of diseases involving immune suppression, cellular aging, DNA damage, and apoptosis [7,13,14]. *Pyracantha angustifolia* is a perennial flowering plant in the Rosaceae family, which is native to China but has been introduced to North America and Australia. This plant is traditionally used as a medicine to treat a range of diseases of the stomach as well as to improve digestion, blood circulation, diarrhea, dysentery, and hemostasis. However, the physiological properties of *P. angustifolia* have not been studied. In the present study, compounds with whitening effects in B16F10 mouse melanoma cells were isolated from *P. angustifolia*, and the anti-melanogenesis and antioxidant effects of its compounds were examined.

## 2. Materials and Methods

### 2.1. Plant Materials

The leaves, twigs, and fruits of *P. angustifolia* were gleaned from Nambu Forest in Seoul National University, Beagwoon Mountain, Gwangyang city, Jeollanam-do, Korea on January 2017. A voucher specimen (SCNUP 22) was deposited in the Laboratory of Pharmacognosy, College of Pharmacy, Sunchon National University, Suncheon-si, Jeollanam-do, Korea.

### 2.2. Extraction and Isolation

The dried leaves (PAL), twigs (PAT), and fruits (PAF) of *P. angustifolia* (10 g each) were pulverized. Their ethanol extracts were obtained by sonication at room temperature and concentrated. These extracts were used to measure the cell viability and bioactivities. Among them, the leaves (859.7 g) with an anti-melanogenesis effect but no cytotoxicity were extracted with 100% ethanol, three times for 2 h, with sonication, and yielded 146.6 g. This was suspended in distilled water (DW) and partitioned successively with *n*-hexane, chloroform (CHCl_3_), ethyl acetate (EtOAc), and *n*-butanol, resulting in solid residues weighing 4.9 g, 14.8 g, 16.9 g, and 14.6 g, respectively. The CHCl_3_ fraction was separated by open liquid chromatography (LC) using a gradient solvent system in normal phase (NP) LC with 150 g of silica gel and an *n*-hexane:EtOAc ratio of 5:1 to 1:1, to obtain twenty subfractions (C1-20). Compounds **3** (1.3 mg, tR 39.1 min) and **5** (1.1 mg, tR 47.5 min) were obtained by separation using a multi-gradient reverse phase (RP) high-performance liquid chromatography (HPLC) system (YMC Triart, YMC Ltd., Kyoto, Japan, C_18_, 250 × 10 mm, H_2_O:CH_3_CN = 95:5 → 100% CH_3_CN, 2 mL/min) from C9 and C10, respectively. C20 was separated using an RP-HPLC system (YMC Triart, C_18_, 250 × 10 mm, H_2_O:CH_3_CN = 20:80 → 100% CH_3_CN, 2 mL/min), and compounds **6** (1.3 mg, tR 20.0 min) and **7** (1.3 mg, tR 17.2 min) were obtained. The EtOAc fraction was separated using a gradient solvent system in open LC with 200 g of silica gel and a CHCl_3_:MeOH ratio ranging from 10:0 to 100% MeOH to obtain ten subfractions (EA1-10). EA8 was separated by multiple preparative liquid chromatography (MPLC) using a Dispo Pack AT column (SIL-25:40 g, 15 × 25 mm, CHCl_3_:MeOH = 10:1 → 100% MeOH) to obtain ten subfractions (EA8-1-10). Compound **1** (1.1 mg, tR 25.6 min) was yielded from EA8-4 using an RP-HPLC system (YMC Triart, C_18_, 250 × 10 mm, H_2_O:CH_3_CN = 95:5 → 100% CH_3_CN, 2 mL/min). The *n*-butanol fraction was subjected to open LC (silica gel 150 g, CHCl_3_:MeOH *=* 50:40:1 → 10:5:1) to obtain ten subfractions (B1-10). The B5 subfraction was separated using an open LC system (silica gel 35.7 g, CHCl_3_:MeOH:H_2_0 = 50:40:1 → 100% MeOH) to afford thirteen subfractions (B5-1~13). Compounds **4** (3.2 mg, tR 28.0 min), **8** (1.5 mg, tR 17.7 min), **11** (5.9 mg, tR 29.1 min), **13** (2.3 mg, tR 27.2 min), and **14** (1.2 mg, tR 26.0 min) were isolated from B5-7 using the RP-HPLC system (YMC Triart, C_18_, 250 × 10 mm, H_2_O:CH_3_CN = 95:5 → 100% CH_3_CN, 2 mL/min). Compound **15** (1.6 mg, tR 33.9 min) was purified from B5-9 using the RP-HPLC system (YMC Triart, C_18_, 250 × 10 mm, H_2_O:CH_3_CN = 80:20 → 100% CH_3_CN, 2 mL/min). B5-13 was separated using an RP-HPLC system (YMC Triart, C_18_, 250 × 10 mm, H_2_O:CH_3_CN = 95:5 → 100% CH_3_CN, 2 mL/min) to yield four compounds, **2** (6.4 mg, tR35 8 min), **9** (1.1 mg, tR 12.9 min), **10** (3.0 mg, tR 50.0 min), and **12** (1.8 mg, tR 42.0 min).

*p*-hydroxybenzoic acid β-d-glucosylester (HG) (**10**). C_13_H_16_O_8_; white amorphous powder; ^1^H NMR (400 MHz, DMSO-*d*_6_); δ 7.87 (1H, d, *J* = 8.8 Hz, H-2), 7.87 (1H, d, *J* = 8.8 Hz, H-6), 6.85 (1H, d, *J* = 8.7 Hz, H-3), 6.85 (1H, d, *J* = 8.7 Hz, H-5), 5.51 (1H, d, *J* = 7.6 Hz, Glc-1), 3.64 (1H, dd, *J* = 4.4, 10.8 Hz, Glc-6a), 3.46 (1H, dd, *J* = 5.9 Hz, Glc-6b), 3.11-3.29 (4H, m, Glc-2, 3, 4, 5). ^13^C NMR (100 MHz, DMSO-*d*_6_): δ 164.4 (C-7), 162.4 (C-4), 132.0 (C-2), 132.0 (C-6), 119.7 (C-1), 115.4 (C-3), 115.4 (C-5), 94.5 (Glc-1), 77.9 (Glc-3), 76.5 (Glc-5), 72.5 (Glc-2), 69.5 (Glc-4), 60.6 (Glc-6).

Cimidahurinine (CD) (**12**). C_14_H_20_O_8_; yellow amorphous powder; ^1^H NMR (400 MHz, DMSO-*d*_6_); δ 6.94 (1H, s, H-6), 6.68 (1H, s, H-2), 6.68 (1H, s, H-5), 4.61 (1H, d, *J* = 5.6 Hz, Glc-1), 3.68 (1H, dd, *J* = 1.2, 5.2 Hz, Glc-6a), 3.50 (2H, dd, *J* = 5.6, 10.4 Hz, H-8), 3.42 (1H, m, Glc-6b), 3.27 (1H, m, Glc-5), 3.26 (1H, m, Glc-2), 3.26 (1H, m, Glc-3), 3.15 (1H, d, *J* = 4.8 Hz, Glc-4), 3.57 (2H, t, *J* = 7.4 Hz, H-7). ^13^C NMR (100 MHz, DMSO-*d*_6_): δ 145.1 (C-3), 145.1 (C-4), 130.3 (C-1), 123.2 (C-6), 117.4 (C-5), 115.5 (C-2), 102.4 (Glc-1), 77.2 (Glc-5), 75.9 (Glc-3), 73.4 (Glc-2), 69.8 (Glc-4), 62.4 (C-8), 60.8 (Glc-6), 38.5 (C-7).

### 2.3. Anti-Melanogenesis Assay

#### 2.3.1. Cell Culture and Viability Assay

B16F10 mouse melanoma cells were purchased from the Korean Cell Line Bank (KCLB) (Seoul, Korea). The cells were cultured at 37 °C in high-glucose dulbecco’s modified eagle medium (DMEM), supplemented with 10% fetal bovine serum (FBS) (HyClone, Logan, UT, USA), 100 IU/mL penicillin, and 100 mg/mL streptomycin (HyClone, Logan, UT, USA) in a humidified atmosphere containing 95% air and 5% CO_2_. Extracts of each fraction, and compounds for the anti-melanogenesis assay of the B16F10 cells, were dissolved in DMSO. The cytotoxicity of *P. angustifolia* was assessed by quantifying the cell viabilities in the presence of each sample. The cells (1 × 10^4^ cells/well) were seeded and incubated for 24 h in DMEM supplemented with 10% FBS. The culture medium was replaced with serum-free DMEM, and treated with the compounds at different concentrations for 72 h. After removing the serum-free DMEM, the cells were treated with an MTT solution (0.5 mg/mL) and incubated for 4 h. After replacing the medium with DMSO, the number of live cells was measured at 570 nm on a microplate reader (BioTek Instruments, Winooski, VT, USA).

#### 2.3.2. Melanin Content Assay

The intracellular melanin content was determined using a slight modification of the procedure described elsewhere [15]. Briefly, B16F10 cells were seeded in a 24-well plate with 2 × 10^4^ cells/well and incubated for 24 h in DMEM containing 10% FBS. The cells were pretreated with the sample for 72 h, washed with PBS, and dissolved in 1N NaOH containing 10% DMSO by boiling at 80 °C for 1 h. The cell lysates were centrifuged at 14,000 rpm for 10 min. The absorbance of the supernatant was measured at 490 nm. The melanin content is expressed as a percentage of the control. Arbutin (500 μg/mL) was used as a positive control.

#### 2.3.3. Cellular TYR Activity Assay

The cellular TYR activity was measured as the l-DOPA oxidase activity using a slight modification of a previously described method [16,17]. The B16F10 cells were seeded and incubated for 24 h in a 24-well plate in the presence or absence of the sample for 72 h. After treatment, the cells were washed with cold PBS and lysed with a lysis buffer (0.1 M sodium phosphate buffer, 1% Triton X-100 and 0.1 mM PMSF). The cell lysates were centrifuged at 10,000 rpm for 10 min. The supernatant was used as the cellular TYR solution. The reaction mixture, containing 40 μL of cell lysate and 100 μL of l-DOPA (2 mg/mL), was incubated at 37 °C for 1 h and the level of dopachrome formation was measured spectrophotometrically at 490 nm. RIPA buffer was used as the control. The TYR activity was calculated as a percentage of the control; arbutin (500 μg/mL) was used as the positive control.

#### 2.3.4. Western Blot Analysis

The protein expression of TYRP-1 and TYRP-2 was measured by western blot analysis. The pretreated and stimulated B16F10 cells were washed with cold PBS and harvested using a cell scraper. The whole cell lysates were extracted with a protein extraction kit (InTRON Biotechnology, Seongnam-si, Korea). Equal amounts of protein were separated by 10% SDS-PAGE and transferred to polyvinylidene difluoride (PVDF) membranes. The membrane was blocked with 5% skim milk in a plain buffer (20 mM Tris pH 7.4 and 136 mM NaCl) at room temperature for 1 h, and incubated overnight with the primary antibodies TYRP-1 and TYRP-2 (SantaCruz Biotech, Santa Cruz, CA, USA) at 4 °C. The membrane was then incubated with a 500-fold dilution of specific secondary HRP-conjugated antibody (Thermo Fisher Sci. Waltham, MA, USA) at RT for 1 h, and the immunoreactive bands were visualized using an enhanced chemiluminescence (ECL) assay kit according to the manufacturer’s instructions.

#### 2.3.5. ROS Assay

Intracellular ROS production was assessed with an oxidant-sensitive fluorescent probe, dichlorofluorescin diacetate (DCFH-DA). B16F10 melanoma cells were seeded at 1 × 10^4^ cells/well and incubated with various concentrations of HG and CH for 24 h. The cells were treated with tert-butyl hydroperoxide (*t*-BHP) (400 μM) for 2 h to induce ROS production, and then incubated with DCFH-DA (25 μM) for 30 min. The fluorescence intensities were measured at an excitation wavelength of 485 nm and an emission wavelength of 535 nm on a fluorescence microplate reader (Berthold Technologies GmbH & Co, KG US, Bad Wildbad, Germany).

### 2.4. Antioxidant Assay

#### 2.4.1. Preparation of Sample 

Extracts of the leaves, twigs and fruits, as well as each fractions of *n*-hexane, CHCl_3_, EtOAc, and *n*-butanol were dried and dissolved in ethanol. The compounds isolated from each fractions were dissolved in DMSO and used to assess antioxidant activity.

#### 2.4.2. DPPH Assay

The radical-scavenging effect of 1,1-diphenyl-β-picrylhydrazine (DPPH) (Sigma-Aldrich, St Louis, MO, USA) was measured using the method reported elsewhere, with a slight modification [18]. A 100 μL quantity of the sample of DPPH solution (0.2 mM) was added to a 100 μL sample on a 96-well plate, mixed for five seconds, and reacted for 30 min in the shade. The absorbance was measured at 517 nm using a microplate spectrophotometer (Epoch, Biotek Instruments, Inc., Winooski, VT, USA); ascorbic acid (100 μg/mL) (Sigma-Aldrich Co.) was used as the positive control.

#### 2.4.3. ABTS Assay

The 2,2-azino-bis(3-ethylbenzothiazoline-6-sulfonic acid diammonium salt (ABTS), Sigma-Aldrich, Co.) radical-scavenging activity was measured using a slight modification of the method reported by Proestos [19]. 2,2′-azobis (2-aminopropane) dihydrochloride (7 mM) (Sigma-Aldrich, Co.) was mixed with 2.45 mM ABTS and then reacted for 16 h at 4 °C. A 50 μL quantity of the sample and 100 μL of the ABTS solution were reacted at 23 °C for 20 min after adding them to a 96-well plate. The absorbance was measured at 734 nm; ascorbic acid (100 μg/mL) was used as the positive control.

### 2.5. Statistical Analysis

All measurements were conducted independently and at least in triplicate. Data are expressed as the mean ± SD. The significant differences between the control and the HG and CH groups were determined using a Student’s *t*-test at *p* < 0.05.

## 3. Results

### 3.1. Anti-Melanogenesis and Antioxidant Effects of Extract

Hyperpigmentation is the overproduction of melanin, and a major factor determining skin color, which causes visible skin pigmentation disorders, such as albinism, melisma, freckles, moles, and lentigo [3,20]. Therefore, skin-whitening agents are applied to treat pigmentation and its related diseases. This study examined the anti-melanogenesis and antioxidant activities of each part of the extracts from *P. angustifolia.* To investigate the anti-melanogenesis activity, the levels of melanin production and TYR activity were examined in B16F10 cells. The DPPH and ABTS radical activities were assessed for their antioxidant activities. For sample preparation, the dried parts of *P. angustifolia*, such as leaves, twigs, and fruits, were extracted with 100% EtOH. The anti-melanogenesis activities of the three extracts were evaluated by studying the inhibition of melanin production and TYR activity in B16F10 cells. Three extracts of the leaves, twigs, and fruits suppressed melanin production at 100 and 250 μg/mL (Figure 1A). Among the three extracts, the leaf extract inhibited TYR activity most strongly, at 100 and 250 μg/mL (Figure 1B). The effects of the extracts, and their effect on the viability of mouse B16F10 melanoma cells, were assessed using an MTT assay to determine the non-toxic concentration of the extracts. None of the extracts had any cytotoxic effects on B16F10 cells after treatment with 100 and 250 μg/mL for 72 h, except 250 μg/mL of the twig extract (Figure 1C). Subsequent experiments were conducted with the leaf extract with no cytotoxicity. DPPH and ABTS assays were conducted to examine the effects of each part of *P. angustifolia* on antioxidant activity. All the extracts exhibited strong DPPH radical-scavenging activity (Figure 1E), but PAL and PAT exhibited strong ABTS radical-scavenging activity (Figure 1D).

### 3.2. Bioactivity-Guided Isolation of Active Phytochemicals from P. angustifolia

The ethanol extracts of PAL, PAT, and PAF were assessed for their anti-melanogenesis activities in B16F10 cells. Among them, PAL suppressed the melanin contents significantly (by 32.4%) at a concentration of 100 μg/mL without cytotoxicity (Figure 1A), whereas PAT showed weak cytotoxicity. Therefore, potent PAL was selected for the isolation of active phytochemicals. A large quantity of PAL extract was partitioned with *n*-hexane, CHCl_3_, EtOAc, *n*-butanol, and aqueous residue, depending on the solvent polarity. All fractions were assessed to determine their effects on the inhibition of melanin production at 100 μg/mL (Figure 2A). The *n*-hexane, CHCl_3_, EtOAc, and *n*-butanol fractions showed 28.4, 35.3, 25.8, and 27.3% inhibition, respectively, without cytotoxicity (Figure 2C). Among them, the *n*-hexane and CHCl_3_ fractions potently suppressed the TYR activity (Figure 2B). In addition, the *n*-butanol fraction showed the strongest DPPH radical-scavenging activity of 69.9% at 100 μg/mL (Figure 2D). In accordance with the ABTS-scavenging activity, the EtOAc and *n*-butanol fractions had higher antioxidant activity than the other fractions (Figure 2C). Therefore, three fractions were used to isolate the bioactive phytochemicals, except for the *n*-hexane fraction that contained a quantity of lipid-like phytosterols (Figure 3). Fifteen compounds, four from the CHCl_3_ fraction, one from the EtOAc fraction, and ten from the *n*-butanol fraction, were isolated by repeated silica gel column chromatography, MPLC, and RP-HPLC, respectively. These compounds were identified as (±)-epicatechin (**1**), cosmosiin (**2**), 9-hydroxyeriobofuran (**3**), 5,7-dihyroxychromone 7-β-d-glucoside (**4**), 2-oxopomolic acid (**5**), pomolic acid (6), ursolic acid (**7**), isopropyl β-d-glucoside (**8**), arbutin (**9**), *p*-hydroxybenzoic acid β-d-glucosyl ester (**10**), roseoside (**11**), cimidahurinine (**12**), 3-(β-d-glucopyranosyloxy)-1-(4-hydroxy-3,5-dimethoxyphenyl)-1-propanone (**13**), dihydrosyringin (**14**), and cinnamoyl glucoside (**15**) by a comparison of the measured spectroscopic (NMR and MS) data with published data (Figure 4) [21,22,23,24,25,26,27,28,29,30,31,32,33,34,35].

### 3.3. Anti-Melanogenesis and Antioxidant Effects of Seven Compounds Isolated from P. angustifolia Fraction

Among the fifteen phytochemicals, the anti-melanogenesis effects of compounds **1**–**3**, **5**–**7**, **9**, and **11** are reported elsewhere. In particular, arbutin, a representative whitening material, was used as a positive control in this study [36,37,38,39,40]. With the exception of these compounds, seven compounds isolated from the *n*-butanol fraction were assessed for their anti-melanogenesis and antioxidant effects. To examine the anti-melanogenesis activity, the levels of melanin production and TYR activity in the pretreated cells were measured. As a result, HG and CH showed strong activities in melanin production (Figure 5A) and TYR activity (Figure 5B) without cytotoxicity (Figure 5C). The ABTS and DPPH radical-scavenging activities were examined, to determine the antioxidant effects of these compounds. Among these compounds, HG and CH exhibited strong DPPH (Figure 5D) and ABTS (Figure 5E) scavenging activity, respectively.

### 3.4. Effects of HG and CH on TYRP-1 and TYRP-2 Expression

TYRP-1 and TYRP-2 are critical enzymes that influence the quantity and quality of melanin [12]. Among the fifteen compounds isolated from PAL, TYRP-1, and TYRP-2, the levels of HG and CH protein expression, which were not previously known to have an anti-melanogenesis effect, were determined by western blot analysis using the specific antibodies. As shown in Figure 6, HG and CH inhibited the expression of these proteins at 10 and 100 μM (Figure 6).

### 3.5. Effect of HG and CH on the Intracellular ROS Production

Melanin synthesis includes oxidation processes and superoxide anion and hydrogen peroxide generation, which are important in ROS production and hyperpigmentation. The inhibitory effects of HG and CH on ROS production in *t*-BHP-treated B16F10 melanoma cells were examined. ROS levels were higher in *t*-BHP-treated cells than in untreated cells. Pretreatment of HG and CH at 10 and 100 μM significantly suppressed ROS production (Figure 7).

## 4. Discussion

The demand for natural pigment reducers for pharmaceutical and cosmetic applications has increased rapidly [4]. Recently, natural products have been considered effective sources of bioactive compounds in the development of oxidation-reducing and hypopigmenting agents. This study explored potential therapeutic components from natural products. This study evaluated the applicability of *P. angustifolia* by bioactivity-guided fractionation to find natural therapeutic products that can reduce hyperpigmentation.

Melanin is a dark pigment that is synthesized in melanocytes by the oxidation of l-tyrosine from external stimuli, such as UV [5,6,7,8,9]. Although the primary role of melanin is the protection of skin tissue from UV irradiation, age spots and freckles can form as a result of the excessive production of melanin [3,4,5,6,7,8,9,10,11,12,13]. Considerable efforts have been made to develop products that reduce melanin synthesis for use as whitening constituents [23,29,36,37,38,39,40,41,42]. TYR is an oxidase and is the rate-limiting enzyme in the production of melanin. This enzyme is a copper-containing enzyme present in plant and animal tissues and is involved mainly in two distinct reactions of melanin synthesis that catalyze the production of melanin and other pigments from tyrosine by oxidation [8,9]. Oxidative stress, which is defined as the production of toxic free-radical species, and an imbalance between pro-oxidant and antioxidant homeostasis, is associated with cellular skin aging [4]. ABTS and DPPH assays are widely used to investigate the antioxidant capacities of natural products [4]. Both assays are associated mainly with the hydrogen-donating or proton-radical-scavenging capacities of the natural products [18,19]. The hydroxyl radical-scavenging properties are involved in two mechanisms: the inhibition of hydroxyl radical production from hydrogen peroxide by binding with metal ions, and direct single electron transfer to the generated radical. Melanin synthesis involves oxidation reactions, and inhibiting oxidative damage is important in the prevention of hyperpigmentation [3,4]. Mouse B16F10 skin melanoma cells were used to elucidate the anti-melanogenesis activity. First, the leaves, twigs, and fruits of *P. anqustifolia* were extracted with EtOH, and the levels of melanin production and TYR activity in B16F10 cells were then investigated. The results showed that all extracts of each part inhibited melanin production (Figure 1A) and TYR activity (Figure 1B) at 100 and 250 μg/mL. On the other hand, none of the extracts exhibited cytotoxicity except for the twig extract at 250 μg/mL (Figure 1C). Three extracts, PAL, PAT, and PAF, were assessed for their radical-scavenging ability, to determine their antioxidant potential. Among the three extracts, PAL and PAT exhibited strong ABTS and DPPH radical-scavenging activities at 100 and 250 μg/mL (Figure 1D). Subsequent experiments were performed with PAL, which showed the most stable and strongest activity in inhibiting melanin production and the activity of TYR. In addition, PAL had a significant effect on the radical-scavenging activities of ABTS and DPPH. The inhibitory effects on melanin production, TYR activity, ABTS, and DPPH radical-scavenging activities were assessed by melanogenesis and oxidant activity-guided fractionation to partition the leaf extract using *n*-hexane, ethyl acetate, *n*-butanol, and H_2_O (Figure 2). To discover the anti-melanogenesis fraction, the fractions were tested to determine if they could inhibit melanin production and the TYR activity in B16F10 cells. Four fractions inhibited melanin production, with the CHCl_3_ fraction exhibiting the strongest inhibition of TYR activity. The DPPH and ABTS radical-scavenging activities were measured to examine the antioxidant fraction. Among the fractions, the EtOAc and *n*-butanol fractions exhibited strong ABTS and DPPH radical-scavenging activities, respectively. Therefore, the CHCl_3_, EtOAc, and *n*-butanol fractions were used to isolate the bioactive phytochemicals. Fifteen compounds, four from the CHCl_3_ fraction, one from the EtOAc fraction, and ten from the *n*-butanol fraction, were isolated and their chemical structures were determined (Figure 4). Among the fifteen phytochemicals, the anti-melanogenesis effects of the compounds are reported elsewhere, as representative whitening materials [36,37,38,39,40]. In the present study, seven compounds isolated from the *n*-butanol fraction were assessed. To search for anti-melanogenesis and antioxidant phytochemicals, this study examined whether the seven compounds (**1**–**7**) isolated from the *n*-butanol fraction could inhibit melanin production and TYR activity. Of the seven compounds, compounds **10** and **12** inhibited melanin production (Figure 5A) and TYR activity (Figure 5B) with no cytotoxicity (Figure 5C). These two compounds were identified as *p*-hydroxybenzoic acid β-d-glucosyl ester (HG) and cimidahurinine (CH). The antioxidant activity of these two compounds, HG and CH, exhibited strong DPPH and ABTS radical-scavenging activity, respectively (Figure 5D). TYR, TYRP-1, and TYRP-2 comprise the TYR family of proteins. TYRP-1 is a melanocyte-specific gene product involved in melanin synthesis and is involved in stabilizing the TYR protein and modulating its catalytic activity. Moreover, TYRP-1 is also involved in maintaining the melanosome structure and affects melanocyte proliferation and melanocyte cell death. TYRP-2 acts as an l-DOPAchrome to produce DHICA via eumelanin synthesis [20,43]. The inhibitory effects of HG and CH on TYRP-1 and TYRP-2 expression were investigated by western blot analysis to confirm the inhibitory relationship between melanin production and the antioxidant effects. These results showed that HG and CH inhibited TYRP-1 and TYRP-2 at 10 and 100 μM in B16F10 cells (Figure 6). Moreover, oxidative stress is the major cause of ROS production, induces disruption of melanocytes homeostasis and triggers melanin synthesis [16]. The inhibitory activities of HG and CH on ROS generation were determined using a ROS-specific probe, DCFH-DA. Our results shown that HG and CH suppressed ROS generation in *t*-BHP-treated B16F10 cells (Figure 7). These results suggest that the hypopigmentation of *P. angustifolia* containing HG and CH is due to the inhibition of radicals produced by oxidation. TYRP-1 expression is regulated by the microphthalmia-associated transcription factor (MITF) [1,43,44]. Moreover, cAMP induces the expression of TYR, TYRP-1, and TYRP-2. The biological activity of cAMP is mediated by cAMP-dependent protein kinase A (PKA), which results in the activation of cAMP-responsive element binding protein (CREB) [1,43]. Further mechanistic studies on the down-regulation of HG and CH in oxidative-stress-induced signaling will be necessary to confirm their potential therapeutic applications in the protection of hyperpigmentation.

## 5. Conclusions

The leaf extract of *P. angustifolia* exhibited anti-melanogenesis and antioxidant effects. HG and CH, which are phytochemicals isolated from *P. angustifolia* using bioactivity-guided fractionation, suppressed melanin generation and TYR activity, and down-regulated the activation of TYRP-1 and 2 in B16F10 cells. Moreover, these two compounds exhibited DPPH and ABTS radical-scavenging activities. Overall, *P. angustifolia*-containing active compounds, such as HG and CH, could be a useful candidate for the production of hypopigmenting agents.

## Figures and Tables

**Figure 1 antioxidants-09-00258-f001:**
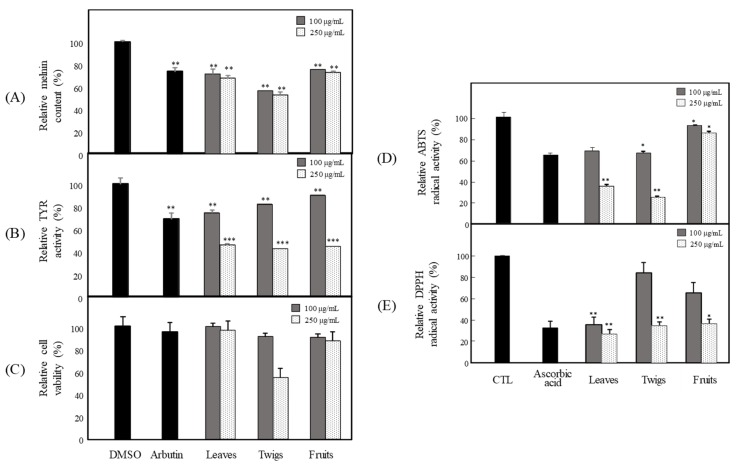
Anti-melanogenesis and antioxidant effects of *P. angustifolia* leaves (PAL), twigs (PAT) and fruits (PAF) extracts. (**A**) Melanin content, (**B**) tyrosinase (TYR) activity (**C**) cytotoxicity in B16F10. (**D**) ABTS and (**E**) DPPH radical-scavenging activities. The data are expressed as the mean ± SD (*n* = 3) of three individual experiments. * *p* < 0.05, ** *p* < 0.01 and *** *p* < 0.001, compared with control.

**Figure 2 antioxidants-09-00258-f002:**
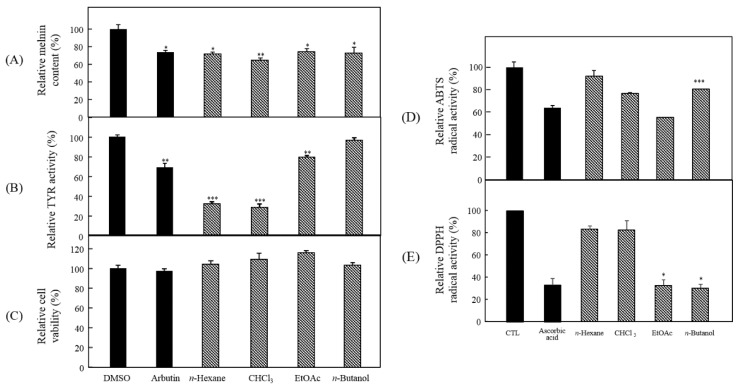
Anti-melanogenesis and antioxidant effects of each fraction of PAL. (**A**) Melanin content, (**B**) TYR activity (**C**) cytotoxicity in B16F10. (**D**) ABTS and (E) DPPH radical-scavenging activities. The data are expressed as the mean ± SD (*n* = 3) of three individual experiments. * *p* < 0.05, ** *p* < 0.01 and *** *p* < 0.001, compared with control.

**Figure 3 antioxidants-09-00258-f003:**
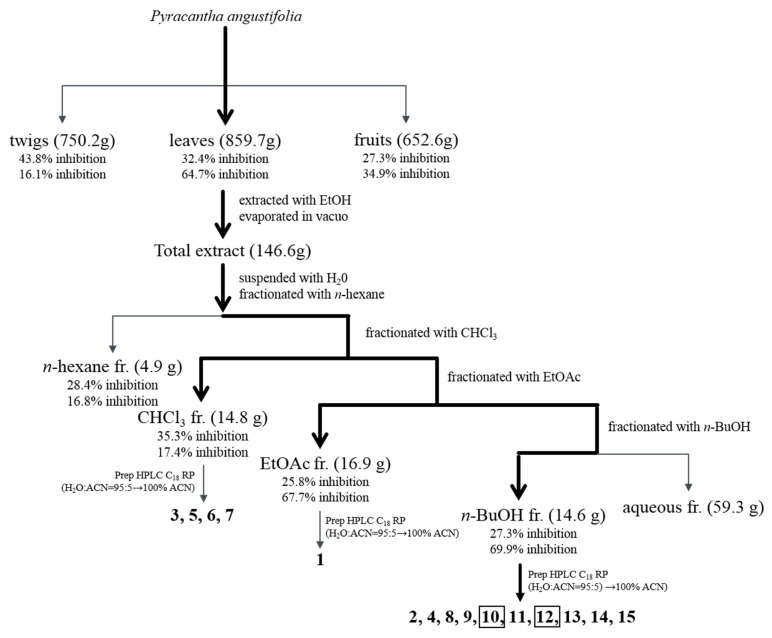
Schematic representation of the isolation of compounds 10 (HG) and 12 (CH) from PAL using bioactivity-guided fractionation. Bioactivity-guided fractionation of PAL was performed as shown in the schematic representation and resulted in the isolation and identification of compounds 10 and 12. Fractionation was guided by assessing the inhibitory effect of HG and CH on melanin content without cytotoxicity and DPPH radical activity at 100 μg/mL. At each level of fractionation, all the fractions generated were tested simultaneously and were compared to the crude extract.

**Figure 4 antioxidants-09-00258-f004:**
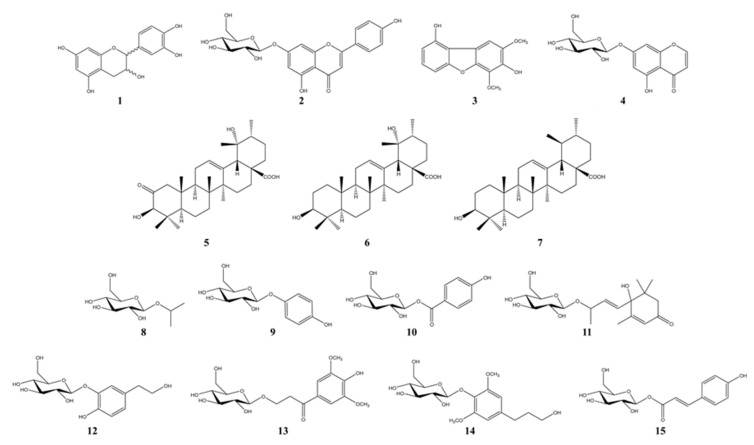
Chemical structures of compounds isolated from PAL.

**Figure 5 antioxidants-09-00258-f005:**
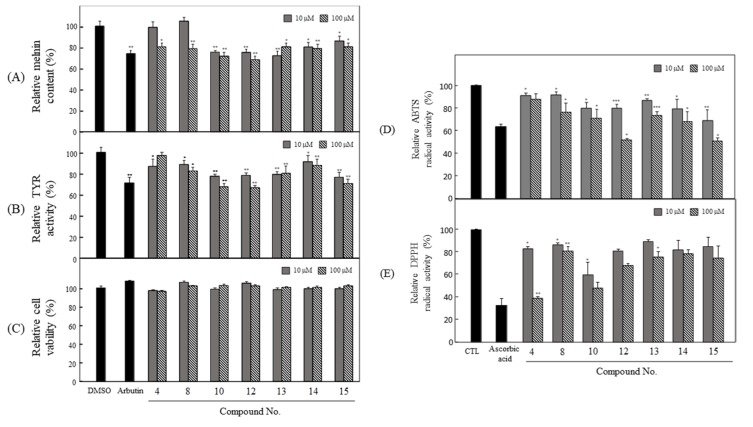
Anti-melanogenesis and antioxidant effects of seven compounds isolated from *n*-butanol fraction of PAL. (**A**) Melanin content, (**B**) TYR activity (**C**) cytotoxicity in B16F10. (**D**) ABTS and (**E**) DPPH radical-scavenging activities. The data are expressed as the mean ± SD (*n* = 3) of three individual experiments. * *p* < 0.05, ** *p* < 0.01 and *** *p* < 0.001, compared with control.

**Figure 6 antioxidants-09-00258-f006:**
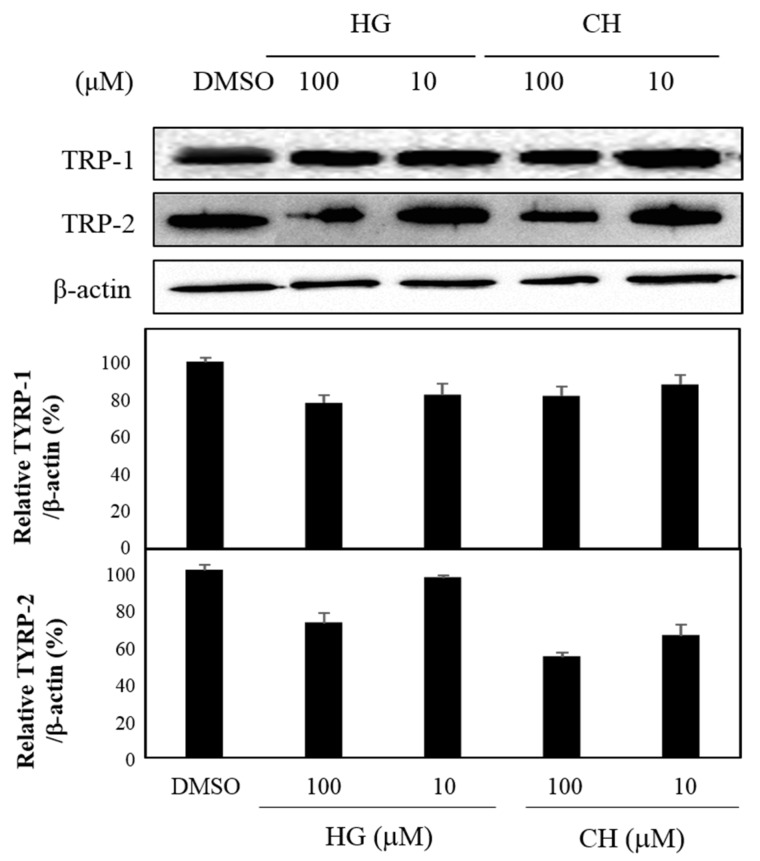
Effects on TYRP-1 and TYRP-2 expression of HG and CH. The data are expressed as the mean ± SD (*n* = 3) of three individual experiments.

**Figure 7 antioxidants-09-00258-f007:**
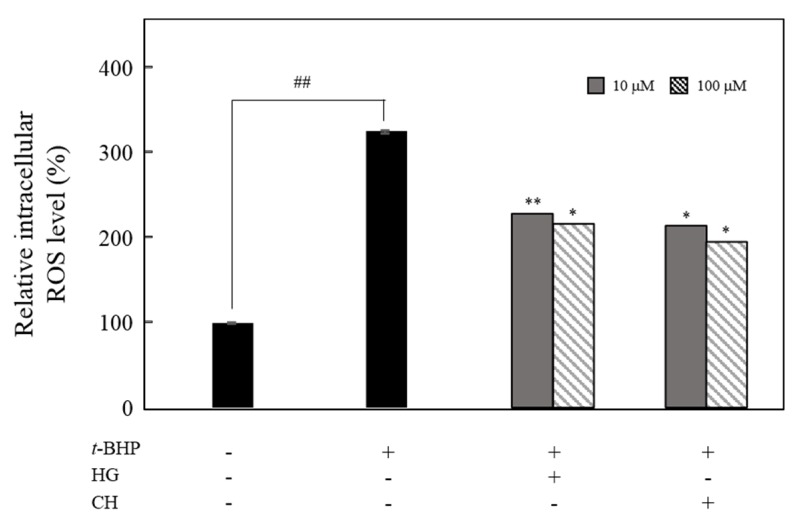
Effect of HG and CH on intracellular ROS levels in *t*-BHP-treated B16F10 mouse melanoma cells. The data are expressed as the mean ± SD (*n* = 3) of three individual experiments. ^##^
*p* < 0.01 vs. non-treated group; * *p* < 0.05 and ** *p* < 0.01 vs. *t*-BHP-treated group.

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
