# Peer review of "p*-Hydroxybenzoic Acid β-d-Glucosyl Ester and Cimidahurinine with Antimelanogenesis and Antioxidant Effects from *Pyracantha angustifolia* via Bioactivity-Guided Fractionation"

_antioxidants, 2020, doi:10.3390/antiox9030258_

Round 1

Reviewer 1 Report

English spelling should be checked.

A correlation between the melanin content and the antioxidative parameters should be provided.

Author Response

Thank you very much for your kind review.

We looking forward to accept our manuscript.

Sincerely yours.

Reviewer 2 Report

The manuscript submitted contains interesting information, perfectly inside the scope of the Journal, but it needs some improvement before acceptation. The following points should be addressed.

Abstract

“Among the seven compounds, p-hydroxybenzoic acid b-d-glucosylester (HG), and cimidahurinine (CD) had strong inhibitory effects of melanin production, TYR activity…….”

The second compound is sometimes abbreviated as CH and sometimes as CD throughout the manuscript. Please take only one choice to avoid confusion.

Introduction

The known biological effects of this plant are stated 3 times (introduction, lines 53-54, results, lines 164-165, discussion, lines 252-253). It is found the sentence “P. anqustifolia has been used traditionally for treating various diseases involving the stomach and improves digestion, blood circulation, diarrhea, dysentery, and hemostasis”. However, there are no references at all. (a) References are needed. (b) Avoid reiteration.

Methods

Line 65: The dried leaves (PAL), twigs (PAT), and fruits (PAF) of P. angustifolia (each 10 g) were pulverized, and their methanol extracts were obtained by sonication at room temperature and concentration. The Methanol extraction.is never presented in results, and there is not comment about it. Delete, as it never used. In turn, there is no data or comments about fruits extraction, with any solvent, ethanol included.

Opposite, at line 209: it is stated that “NMR and MS) data with published papers (Figure 4) [16-32]”. There is no comment in methods about procedures for these techniques and how the identification of compounds was achieved. Thus, the confidence in a correct identification is not assessed. Some general details, or supplementary material related to product identification, at least for compounds 10 and 12, are needed.  

Line 105; The concentration of MSH should be given. In turn, for future studies, MSH is not stable during 72h. Stable analogs to MSH would be more convenient

Line 144; ascorbic acid. Concentration is also required

Line 149-150: Seven millimoles 2,2'-Azobis (2-aminopropane) dihydrochloride (Sigma-Aldrich, Co.) was mixed with 2.45 mM ABTS and then reacted for 16 h at 23 °C.

The aim of the pre-mixing is confusing…… What for? Just ABTS is enough as stable free radical

Minor points

Lines 30-31: melanocytes are melanin-producing cells found in the skin, hair follicles, eyes, inner ear, but they are not confirmed in bones, heart, and human brain. Please, modify the sentence. By the way, some references are not really convenient.

Lines 39-40: TYRP-1 and TYRP-2 are present in melanosome and play an essential role in catalyzing eumelanin-producing reactions. Later, at line 239, it is used the term “critical”.

TRP1 and TRP2 are important enzymes, but they are not essential or critical. Eumelanin is formed once tyrosinase is active and dopaquinone is formed, even in the complete absence of TRP and TRP2. Replace these words. I cannot recommend the appropriate references for that (it is not my rule as reviewer), but anyway some references of this manuscript are quite poor. If desired, look for Aroca et al., Olivares et al in the 90’s for further information about TRP1 and TRP2 fucntions.

Line 156: the control and EDG and DC groups  ………… Define EDG and DC groups

Lines 172-173: “Compared to the activity of arbutin, the positive control, the activities were stronger in the order of twigs and leaves”. The expression “stronger in the order” is unclear. Please, modify

Line 220: DPPH radical activity at 100 g/mL. Something is missing, probably m.

Author Response

(The authors gave the same response as above.)

Reviewer 3 Report

Although the manuscript seems to meet the criteria for publication in Antioxidants, the authors should explain and answer some important comments. Their explanation should be included in the corrected manuscript.

  • English must be improved : for example datas or data ?! please check the entire text carefully.
  • What was the solvent used for HG and CD compounds in DPPH, ABTS, Melanin Content Assay, Cellular TYR Activity Assay and Western Blot Analysis?
  • Did the authors check the effects of n-butanol, and/or the other solvents present in extracts on bioactivities of their novel compounds?

Author Response

(The authors gave the same response as above.)

Round 2

Reviewer 1 Report

I appreciate the authors response and revisions. 

Reviewer 2 Report

Authors performed an excellent job with the original study, and then they have performed a good job to address all my queries. The think that the manuscript has been improved. I have observed that some data have been complemented, corrected or omitted: MSH is not used, conditions of some experiments have been corrected, some unknown groups has disappeared and so on. The addition of NMR data on compounds 10 and 12 is particularly important for supporting the manuscript. I am surprised about the absence of no references about P. angustifolia. Perhaps other beneficial effects on health promised by traditions are waiting justification by a scientific study.

About the last sentence of the reply letter, permit me to say that this is the role of the reviewers, so that I am glad for the final outcome. I also would like to thank authors for the explanation concerning the mechanism of ABTS radicals by pre-mixing with AAPH. I have learnt something about it, as I have read so many papers using just ABTS as direct radical. I am advised for the future.      

Reviewer 3 Report

The manuscript in this new corrected form could be published in the Journal.